# Sustainable Cosmetics: Valorisation of Kiwi (*Actinidia deliciosa*) By-Products by Their Incorporation into a Moisturising Cream

Sandra M. Gomes [1,2], Rita Miranda [3] and Lúcia Santos [1,2,*]

1 LEPABE—Laboratory for Process Engineering, Environment, Biotechnology and Energy, Faculty of Engineering, University of Porto, Rua Dr. Roberto Frias, 4200-465 Porto, Portugal
2 ALiCE—Associate Laboratory in Chemical Engineering, Faculty of Engineering, University of Porto, Rua Dr. Roberto Frias, 4200-465 Porto, Portugal
3 FEUP—Faculty of Engineering, University of Porto, Rua Dr. Roberto Frias, 4200-465 Porto, Portugal
* Correspondence: lsantos@fe.up.pt; Tel.: +351-225-081-682; Fax: +351-225-081-440

**Abstract:** The growing population has intensified food processing, increasing the generation of agro-industrial waste. This waste is rich in bioactive compounds; therefore, it can be valorised by extracting their compounds of biological interest and incorporating them into cosmetic products. In this work, an extract was obtained from kiwi peels and characterised regarding its biological properties and phenolic composition. Results demonstrated that the extract presented antioxidant activity against DPPH and ABTS radicals ($IC_{50}$ values of 244 mg/L and 58 mg/L, respectively) and antibacterial activity against *Staphylococcus aureus*. Catechin and epicatechin (flavonoids), as well as chlorogenic acid (phenolic acid), were the main phenolic compounds identified. Subsequently, the kiwi peel extract was incorporated into cosmetic formulations and their antioxidant properties and stability were evaluated. An increase in the antioxidant activity of the moisturising cream was observed upon the extract's addition. Also, no microorganisms were present in any formulation prepared, attesting to their microbial safety. Finally, the results from the stability analysis revealed that the moisturising creams remained relatively stable for two weeks. These findings suggest that extracts from kiwi peels have the potential to be used as natural additives to produce value-added cosmetic products in a more sustainable manner.

**Keywords:** agro-industrial by-products; kiwi peels; phenolic compounds; natural antioxidants; moisturising creams; sustainability

## 1. Introduction

Nowadays, we are observing a rapid increase in the global population, which results in an escalation in food production. This has resulted in intensified agricultural practices to meet the rising demand. Moreover, although the advancements in food processing technologies may have increased production efficiency, they also led to an increase in the generation of waste and food by-products [1,2], which presents serious environmental and socioeconomic consequences. When food waste is disposed of in landfills, it produces greenhouse gases that are harmful to the environment. Moreover, managing large quantities of waste is challenging at the economic level due to the costs associated with processing solid residues in landfills [3]. Additionally, food waste raises ethical and moral concerns, as millions of people are suffering from hunger worldwide [4]. To tackle these challenges in an effective way, it is imperative to adopt sustainable practices and employ innovative approaches to manage and valorise these by-products.

The kiwifruit-processing industry is expanding, not only in Portugal but also worldwide, where new and improved strategies are continuously being developed. This growth is leading to the generation of considerable amounts of agro-industrial waste, such as kiwi

leaves, peels, and pomace [5]. There are multiple types of kiwi cultivars, selected from wild-growing populations of its plant. They can vary in shape, size, amount of hair, or external colour, and their flesh can have different flavours, colours, textures or compositions [6]. The most commonly known variety of kiwi (Hayward) is derived from *Actinidia deliciosa*. Its vine grows naturally at elevated altitudes, between 600 and 2000 m, and the plant itself can reach around 9 m. In the northern hemisphere, this plant grows new leaves in mid-March, which bloom in early May and the fruit can be harvested in November after the leaves have fallen [7]. Historically, kiwi fruit originated in the temperate and mountainous forests of northern China, dating back to the 12th century [8]. However, due to the vast climate conditions its plant can tolerate, in the early 20th century, the cultivation of this fruit became popular in New Zealand and, since then, started to be commercialised and produced in various countries in the world [6]. In 2019, global kiwi production reached approximately 4.3 million tonnes, while its consumption was estimated to be around 4.2 million tonnes. China is the largest kiwi-producing and -consuming country worldwide, with an estimated production of 2.1 million tonnes and consumption of 2.2 million tonnes, in 2019 [9]. In Portugal, this fruit became common only in the last century, and in 2021, around 55 thousand tonnes of kiwi were produced in the country [10]. The waste products generated are generally discarded and transported to landfills, which has a negative impact on the environment, due to the accumulation of untreated agricultural wastes, leading to pollution and health hazards [11]. Besides, they could be utilised much more efficiently by applying the "zero waste" principle to this area, where applying waste as a novel raw material in different applications and using it to develop innovative products could lead to a decrease in socioeconomic and environmental issues while being an important step towards a circular economy [12,13].

As previously mentioned, the kiwi-processing industry generates various by-products, such as leaves, peels, and pomace. Different studies reported that, in general, fruit peels, compared with their flesh, can present greater amounts of nutrients, in particular antioxidants [14]. In the particular case of kiwi, its peels contain mainly carbohydrates and are very abundant in different minerals such as magnesium, potassium or calcium. Also, the phenolic content in kiwi peels (KP) was shown to be 15 times higher than in its flesh [15]. KP are notorious for being rich in multiple phenolic compounds, such as protocatechuic, caffeic and chlorogenic acids, isoquercetin, catechin, and rutin, among others [16–18]. A summary of the chemical composition of KP is presented in Figure 1, which shows that these underused peels can be an attractive source of multiple bioactive compounds (BACs) [19] that present numerous biological properties, namely antioxidant, antimicrobial, and anti-inflammatory benefits [20].

Therefore, new strategies need to be developed to minimise waste generation and to give a second life to the BACs present in these raw materials. Some research has been done in order to find alternative uses for kiwi pomace [21], but much less has been done on the valorisation of KP. Table 1 summarises the investigation that has been performed on the extraction and identification of the BACs present in KP, and its possible application in products from different industries has also been studied.

As it is possible to observe from Table 1, there is very scarce information on the incorporation of kiwi by-products in the cosmetic industry. Nevertheless, cosmetic formulations and, in particular, skincare products, can offer a more sustainable and greener option for the use of the BACs found in KP. Besides contributing to skin health and appearance, adding polyphenols to cosmetic products can provide photoprotection, anti-ageing and anti-inflammatory benefits [22]. Also, natural products are increasingly becoming more popular and attracting the attention of consumers, due to their biological potential, safety, and effectiveness, which makes this market niche a profitable source for producers.

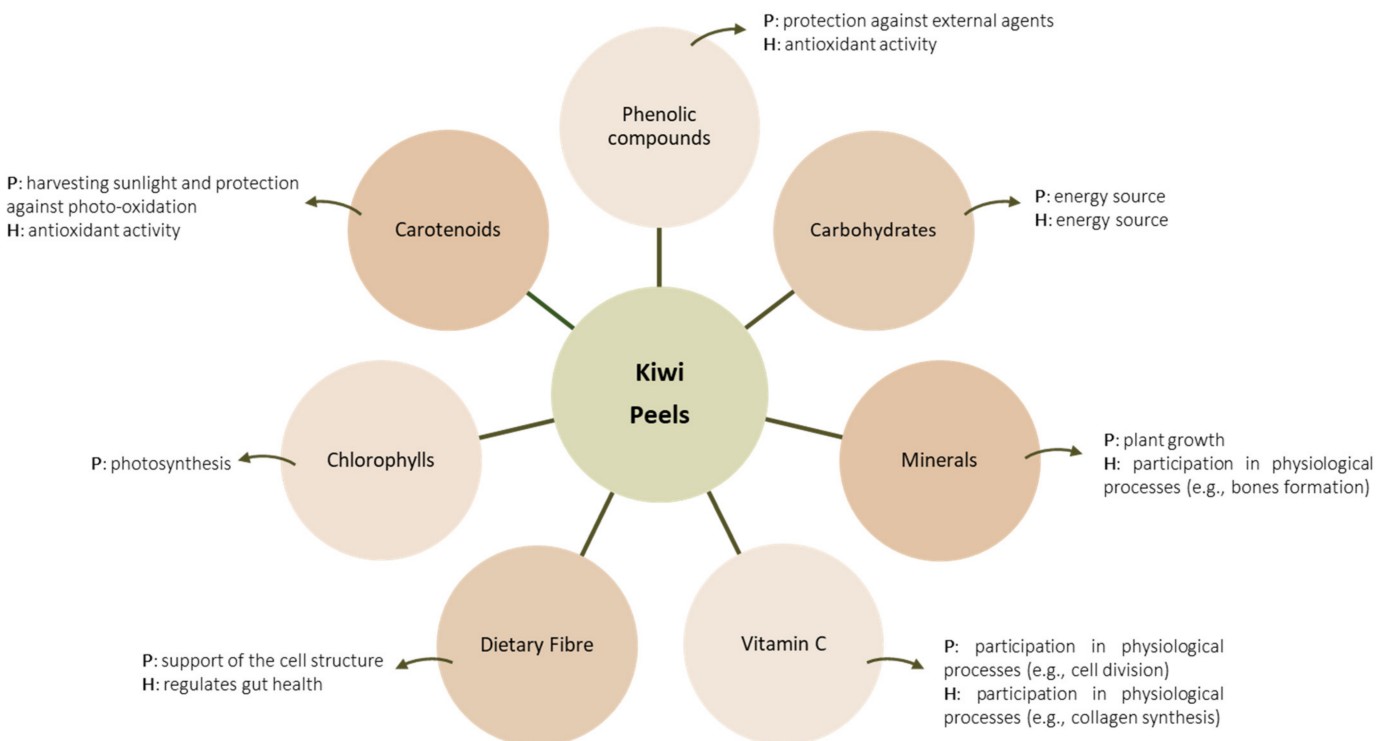

**Figure 1.** Overview on the chemical composition of kiwi peels and the main roles of each constituent in plants (P) and humans (H).

**Table 1.** Studies on the incorporation of kiwi peels on different industries.

| | Product | Objectives | Results | Ref. |
|---|---|---|---|---|
| Food Industry | Flour | Analyse the microbiological and physicochemical properties as well as the BACs of flour made from the KP and bagasse. | Flour made with bagasse presented lower levels of BACs and antioxidant activity compared to flour made with KP. | [23] |
| | Chicken meat emulsion | Obtain antioxidant-rich extracts from KP and study its antioxidant efficacy in vivo in chicken meat emulsions. | KP extract (2.5%) improved the sensory and physicochemical properties of meat model systems and extended their shelf-life. | [24] |
| | Food packaging films | Reutilise KP in the preparation of multifunctional sodium alginate-based food packaging films. | KP extracts were able to bio-reduce silver particles, exhibiting strong antibacterial activity and biocompatibility. They also enhanced the antioxidant and antibacterial activities of sodium alginate films. | [25] |
| Pharmaceutical Industry | ND | Obtain a phenolic-rich extract from KP and explore its capability to modulate inflammatory responses. | The reported findings demonstrated a strong and broad anti-inflammatory profile of the KP extract, which makes it a promising natural ingredient for pharmaceutical formulations, with the potential for prevention and treatment. | [26] |

**Table 1.** *Cont.*

| | Product | Objectives | Results | Ref. |
|---|---|---|---|---|
| Skincare Industry | ND | Summarise the possible cosmetic and medical skincare applications of the different BACs from kiwi by-products, exploring their potential to contribute as active ingredients based on the biological activities reported. | The BACs present in the kiwi by-products contributed to good skin appearance and health, and the addition of these active compounds to the final cosmetic and pharmaceutical products may provide skin-beneficial effects such as anti-ageing, photoprotection, and anti-inflammatory effects, also acting as depigmenting and emollients. | [22] |

ND—not defined; BACs—bioactive compounds; KP—kiwi peels.

Regarding the safety of the KP extracts for humans, no literature reports were found; however, some authors have investigated the cytotoxicity of the extracts [25,27]. In recent years, some allergic reactions to kiwifruit have been reported but the mechanisms behind the reaction is still unclear [28]. Nevertheless, very scarce information is available in the literature about this topic. Therefore, more research is necessary on the toxicity of KP extracts to ensure the consumer's safety when they are added to a new product.

The purpose of this study was to investigate the potential of KP extract regarding its antioxidant and antimicrobial properties, as well as its impact when incorporated into cosmetic products, namely a moisturising cream. The possibility of using this extract as a substitute or to diminish the quantity of synthetic preservatives in skincare formulations was also analysed. To the best of the authors' knowledge, this is the first study to investigate the use of KP extract in cosmetic products, making it a pioneer study in this field. This work aimed to give a solution to the problem of lack of valorisation of the kiwi peels, a by-product with a high nutritional value, that is seemingly forgotten, ending in landfills.

## 2. Materials and Methods

### 2.1. Samples and Reagents

Kiwis were harvested unripened in November 2022, in Viseu, Portugal ($40°66'44''$ N, $7°93'75''$ W), ripening naturally until eaten, in February/March 2023 when the kiwi peels were collected. For extraction and characterisation, the solvents ethanol (Ref. 83813.360, $C_2H_6O$, CAS 64-17-5), acetonitrile (Ref. 45983, $C_2H_3N$, CAS 75-05-8), and methanol (Ref. 20834.291, $C_3H_2O$, CAS 67-56-1) were acquired from VWR (Radnor, PA, USA). For the determination of the total phenolic content and antioxidant properties, 2,2′-azino-bis(3-ethylbenzothiazoline-6-sulfonic acid) (Ref. A1888, $C_{18}H_{24}N_6O_6S_4$, CAS 30931-67-0), 2,2-diphenyl-1-picrylhydrazyl (Ref. D9132, $C_{18}H_{12}N_5O_6$, CAS 1898-66-4), and the Folin–Ciocalteu reagent (Ref. 47641) were obtained from Sigma-Aldrich (St. Louis, MO, USA), while sodium carbonate (Ref. 13418, $CNa_2O_3$, CAS 497-19-8) was purchased from Honeywell (Charlotte, NC, USA). For the microbiological analyses, dimethyl sulfoxide (Ref. 41640, $C_2H_6OS$, CAS 67-68-5) was acquired from Honeywell, sorbic acid (Ref. S1626, $C_6H_8O_2$, CAS 110-44-1) and m-Lauryl Sulfate Broth (Ref. 0734) from Sigma-Aldrich, Plate Count Agar (Ref. 84608.0500) and agar (Ref. J637, CAS 9002-18-0) from VWR, and Rose-Bengal Chloramphenicol Agar (Ref. 1.00467.0500) from Merck (Darmstadt, Germany).

For the quantification of the phenolic compounds, caffeic acid (Ref. C0625, $C_9H_8O_4$, CAS 331-39-5), catechin (Ref. C1251, $C_{15}H_{14}O_6$, CAS 225937-10-0), chlorogenic acid (Ref. C3878, $C_{16}H_{18}O_9$, CAS 327-97-9), epicatechin (Ref. E1753, $C_{15}H_{14}O_6$, CAS 490-46-0), ferulic acid (Ref. PHR1791, $C_{10}H_{10}O_4$, CAS 537-98-4), quercetin (Ref. Q4951, $C_{15}H_{10}O_7$, CAS 117-39-5), and kaempferol (Ref. 60010, $C_{15}H_{10}O_6$, CAS 520-18-3) were obtained from Sigma-Aldrich.

To produce the moisturising creams, betaine (Ref. COR-COSM-00969), coconut oil (Ref. INCOCO-00185), glycerine (Ref. COSM-01216), sweet almond oil (Ref. ACEI-01886), and xanthan gum (Ref. GOMA-00762) were purchased from GranVelada (Zaragoza, Spain), soy lecithin (Ref. L0023) from TCI (Tokyo, Japan) and 2,6-bis(1,1-dimethylethyl)-4-methylphenol (BHT) (Ref. B1378, $C_{15}H_{24}O$, CAS 128-37-0) was acquired from Sigma-Aldrich.

### 2.2. Phenolic Compounds Extraction from Kiwi Peels

Prior to extraction, the kiwi peels (KP) were subjected to a pre-treatment. First, the kiwis were washed and peeled. The excess of water in the peels was removed with a paper towel. Then, the peels were lyophilised for 3 days to ensure the complete removal of the water. Finally, KP were ground using a Coffee Grinder Q5321 (Qilive, Croix, France) until a homogeneous sample was obtained, with a particle size < 0.3 mm.

To extract the phenolic compounds from the KP powder, a solid–liquid extraction method was employed using a Soxhlet apparatus, according to the literature, with some modifications [29]. The extraction was performed for 2 h, using ethanol as solvent in a 1:20 m/V sample-to-solvent ratio. Finally, ethanol was evaporated using a Rotavapor R-200 (BUCHI Laboratories, Switzerland) and a stream of nitrogen of 2 mbar until no variations in the weight of the extract were observed. The extraction yield (EY) was determined using Equation (1):

$$\text{EY (\%)} = (m_{\text{KP extract}}/m_{\text{KP powder}}) \times 100, \tag{1}$$

where $m_{\text{KP extract}}$ refers to the mass of the kiwi peels extract obtained and $m_{\text{KP powder}}$ refers to the mass of the kiwi peels powder initially used for the extraction.

### 2.3. Extract Characterisation
### 2.3.1. Total Phenolic Content

The Folin–Ciocalteu method was used to determine the total phenolic content (TPC) of the KP extract, according to the literature [30].

### 2.3.2. Antioxidant Properties

The antioxidant activity of the extract was investigated by determining its ability to inhibit two different radicals: 2,2-diphenyl-1-picrylhydrazyl (DPPH) and 2,2′-azino-bis(3-ethylbenzothiazoline-6-sulfonic acid) (ABTS). The assays were performed according to the literature, with slight modifications [30]. The KP extract (1.5–8.0 g/L for DPPH and 0.1–2.5 g/L for ABTS) was incubated for 40 min with DPPH solution and 15 min with ABTS solution. Afterwards, the absorbance was analysed at 515 nm for DPPH assay and 734 nm for ABTS assay, with a Thermo GENESYS™ 10UV UV-Vis spectrophotometer (Thermo Fisher Scientific, Waltham, MA, EUA), to calculate the inhibition percentage of the radicals. Finally, the $IC_{50}$ value for each radical was determined.

### 2.3.3. Antibacterial Properties

The antibacterial activity of the KP extract (1 g/mL in 2% DMSO) was studied against *Escherichia coli* (DSM 1103), *Staphylococcus aureus* (335 PF) and *Staphylococcus epidermidis* (DSM 20044-1115-001) using the disk diffusion assay, according to the literature [30]. Sorbic acid (1 g/L) was used as the positive control.

### 2.3.4. Phenolic Composition

The main phenolic compounds present in the KP extract were identified and quantified using high-performance liquid chromatography (HPLC) with a diode array detector (DAD) [30]. To prepare standards and sample solutions, acetonitrile/water/ethanol (2:1:1 *v/v/v*) was used as solvent. Milli-Q water, acidified with 0.5% orthophosphoric acid, and methanol/acetonitrile (80:20 *v/v*) were the eluents used for mobile phase A and mobile phase B, respectively. The volume of injection was 40 μL and the flow rate was set to 0.8 mL/min. The analysis was performed with the following gradient: 90% A + 10% B (0–10 min); 85% A + 15% B (10–25 min); 70% A + 30% B (25–40 min); 50% A + 50% B

(40–50 min); 30% A + 70% B (50–60 min). A calibration curve was prepared for each standard analysed. In the KP extract, the analytes were identified by their retention time (RT) and quantified by the external standard method at their maximum absorption wavelength: 222 nm for catechin and epicatechin; 322 nm for caffeic, chlorogenic and ferulic acids; and 365 nm for kaempferol and quercetin.

### 2.4. Moisturising Creams Production and Characterisation

#### 2.4.1. Formulations Preparation

In order to assess the possibility to use KP extracts as an ingredient of moisturising creams, different oil-in-water (O/W) emulsion formulations were prepared according to the literature [31]. The ingredients of each formulation are described in Table 2. Four different formulations were prepared: the negative control (NC) where no additives were added, the positive control (PC) containing a synthetic antioxidant (BHT), a formulation containing the KP extract that works as a natural antioxidant (KPC), and a formulation containing both BHT and KP extract (Mix). The antioxidant properties, microbial safety and stability of the formulations were analysed to understand the effect of the addition of KP extract to moisturising creams.

**Table 2.** List of ingredients for each moisturising cream formulation.

| | Ingredients | Function | % (*w/w*) | | | |
|---|---|---|---|---|---|---|
| | | | NC | PC | KPC | Mix |
| Aqueous Phase | Water | Solvent | 73.60 | 73.10 | 73.10 | 73.10 |
| | Glycerine | Humectant | 7.60 | 7.60 | 7.60 | 7.60 |
| | Xanthan gum | Thickener | 0.60 | 0.60 | 0.60 | 0.60 |
| Oil Phase | Coconut oil | Emollient | 7.60 | 7.60 | 7.60 | 7.60 |
| | Sweet almond oil | Emollient | 6.80 | 6.80 | 6.80 | 6.80 |
| | Betaine | Emulsifier | 2.70 | 2.70 | 2.70 | 2.70 |
| | Soy lecithin | Emulsifier | 1.10 | 1.10 | 1.10 | 1.10 |
| Additives | BHT | Synthetic antioxidant | - | 0.50 | - | 0.25 |
| | KP extract | Natural antioxidant | - | - | 0.50 | 0.25 |

BHT—2,6-bis(1,1-dimethylethyl)-4-methylphenol; KP—kiwi peels; NC—moisturising cream with no additives (negative control); PC—moisturising cream with a synthetic antioxidant (positive control); KPC—moisturising cream with kiwi peel extract; Mix—moisturising cream with BHT and kiwi peel extract.

#### 2.4.2. Antioxidant Properties

To evaluate the antioxidant properties of the moisturising creams produced, the phenolic compounds were extracted from each formulation. Briefly, 2 g of each formulation were combined with 4 mL of ethanol. Then, the mixture was vortexed for 1 min and placed in an ultrasonic bath for 5 min. This process was repeated three times, after which the mixture was centrifuged (20 min, 3000 rpm). The supernatant was then collected. Finally, the precipitate was combined with another 4 mL of ethanol and the procedure described before was repeated. The antioxidant activity of the supernatants obtained was analysed using DPPH and ABTS assays, as described in Section 2.3.2. These properties were analysed for two weeks.

#### 2.4.3. Microbial Contaminations

The presence of microbial organisms in the moisturising creams was assessed one week after production of the formulations, using Rose Bengal Chloramphenicol Agar (RBC) to detect yeast and moulds and Lauryl Sulphate Agar (LSA) to detect coliform microorganisms, according to the literature [30]. The RCB plates were incubated at 25 °C for 7 days and the LSA plates were incubated at 37 °C for 24 h.

### 2.4.4. Formulations Stability

To analyse the stability of the formulations, both the accelerated physical stability and the pH of the moisturising creams were evaluated right after production and two weeks after production. For the accelerated physical stability evaluation, 2 g of each formulation were centrifuged for 10 min at 4000 rpm and the occurrence of changes in the colour or phase separation was visually analysed. For the pH measurement, 9 mL of water was used to dilute 1 g of each formulation and a pH meter was used to determine the pH value of the solutions, which were under constant agitation.

### 2.5. Statistical Analysis

To perform a statistical analysis of the results, GraphPad Prism 8.0.2 was used to execute a two-way analysis of variance (two-way ANOVA) using Tukey's multiple comparisons test. Results were considered statistically different when *p*-values inferior to 0.05 (with a 95% confidence interval) were obtained.

## 3. Results and Discussion

### 3.1. Characterisation of the Kiwi Peel Extract

To evaluate the potential use of kiwi peel (KP) extracts in cosmetic formulations, an extract was obtained, and its biological properties were assessed. The phenolic compounds were extracted from KP powder using a solid–liquid extraction with a Soxhlet apparatus for 2 h, with a 1:20 m/V sample-to-solvent ratio. The solvent, ethanol, is a Generally Recognised as Safe Substance (GRAS). The GRAS status is extremely important for applications intended for human use to prevent potential toxicity problems. Using this method, an extraction yield of $45.8 \pm 2.8\%$ was achieved and the extract presented a total phenolic content (TPC) of $51.9 \pm 3.7$ mg of gallic acid equivalents (GAE)/g of dried extract. The TPC of KP extracts obtained by solid–liquid extraction techniques was evaluated by other researchers. In one study, where the extraction was performed for 2 h using 50% ethanol and a sample-to-solvent ratio of 1:25 m/V, a TPC close to 20 $mg_{GAE}$/g was achieved [5]. In another study, using 80% acetone in a sample-to-solvent ratio of 1:1 m/V for 10 min to extract the phenolic compounds, a TPC between 2.8 and 4.5 $mg_{GAE}$/g (depending on the origin of the peels) was obtained [18]. Other researchers used methanol to extract the bioactive compounds from KP. The extraction was performed for 1 h (1:20 m/V), the sample was filtered and the residue was extracted again with 80% methanol. These conditions resulted in an extract with a TPC of 12.8 $mg_{GAE}$/g [16]. Finally, the obtention of KP extracts using an ultrasound-assisted method was also studied. The ultrasonic treatment was performed for 1 h using 70% ethanol as solvent (1:20 m/V) and the extract presented a TPC between 20 and 30 $mg_{GAE}$/g, depending on the peels' origin [17]. These results demonstrate that the phenolic content of the KP extracts greatly depends on the extraction technique and conditions and that the method applied in the present study seemed to be more effective in extracting phenolic compounds than the ones utilised in previous studies.

The extract was also characterised regarding its antioxidant and antimicrobial activities, and the obtained results are described in Table 3. Regarding the antioxidant activity, the results demonstrated the ability of KP extract to inhibit both DPPH and ABTS radicals. The extract was found to be more effective in inhibiting ABTS, compared to DPPH, since a higher concentration of extract is required to inhibit DPPH to the same extent as ABTS (higher $IC_{50}$ value for DPPH). The ability of the KP extracts to inhibit these two radicals was also demonstrated by other authors [5,16,17]. However, the $IC_{50}$ values were not presented, making it difficult to compare the results. Compared with other extracts, obtained from plants or agro-industrial waste, such as *Moringa oleifera* leaves, and onion and passion fruit peels, the KP extract obtained in the present work presented a higher $IC_{50}$ value for DPPH than all the other extracts, but a lower $IC_{50}$ value for ABTS when compared to *Moringa oleifera* leaves [29,30].

**Table 3.** Results from the antioxidant and antibacterial properties of the kiwi peel extract.

| Antioxidant Activity | | | Antibacterial Activity | | |
|---|---|---|---|---|---|
| IC$_{50}$ (mg/L) | | | d$_{halo}$ (mm) | | |
| DPPH | ABTS | SA | *E. coli* | *S. aureus* | *S. epidermidis* |
| 243.94 $\pm$ 6.79 | 58.29 $\pm$ 4.00 | ND | ND | 8.0 $\pm$ 0.8 | ND |

IC$_{50}$—concentration of extract necessary to inhibit 50% of the free radical; DPPH—2,2-diphenyl-1-picrylhydrazyl; ABTS—2,2′-azino-bis(3-ethylbenzothiazoline-6-sulfonic acid); SA—sorbic acid (positive control); ND—not detected. The results are expressed as means $\pm$ standard deviations of three independent measurements.

To analyse the antibacterial activity of the KP extract, three different bacteria were used for the assay. More specifically, Gram-negative *E. coli* and Gram-positive *S. aureus* were selected as bacteria models since they are commonly responsible for human infections. Since the extract obtained is intended to be applied in a cosmetic product, more specifically a moisturising cream, a third bacteria was tested, *S. epidermidis*, which is an opportunistic bacterium naturally present in human skin. Observing the results from Table 3, it is possible to conclude that, at the concentration analysed (1 g/L), the KP extract was only able to inhibit the growth of *S. aureus*. In a study where the antimicrobial activity of the KP extract was analysed using *E. coli*, *S. aureus* and *L. monocytogenes*, it was observed that it presented a more potent effect in the inhibition of the growth of *S. aureus*, compared to *E. coli* [16]. This difference in effect can be attributed to the protective outer membrane present in Gram-negative bacteria. This study also demonstrated that bacteria growth inhibition is a dose-dependent mechanism and, therefore, higher concentrations of KP extract can be tested to fully understand its antibacterial potential.

Finally, the phenolic compounds present in the extract were identified and quantified using HPLC. The chromatograms obtained are presented in the Supplementary Material (Figures S1 and S2). Caffeic acid, catechin, chlorogenic acid, epicatechin, ferulic acid, kaempferol and quercetin were the phenolic compounds analysed since the literature reports their presence in KP [16,17]. The results from the quantification of the phenolic compounds present in the extract obtained in the current study are displayed in Table 4. The main phenolics found were catechin, chlorogenic acid and epicatechin, followed by caffeic acid. Ferulic acid, kaempferol and quercetin were not detected. Different factors, such as the extraction method and conditions, can affect the phenolic content of the extract. Also, previous research demonstrated that the phenolic composition of the kiwi peel can greatly vary depending on the fruit origin; not only the same phenolic compound can display a substantial difference in its concentration (e.g., 1.42–356.38 mg/g for catechin), but also the main phenolic compound present is utterly distinct between the different varieties of kiwi [17]. The presence of phenolic compounds such as caffeic acid, catechin, chlorogenic acid and epicatechin may explain the antioxidant properties displayed by the KP extract.

**Table 4.** Phenolic composition of the kiwi peel extract analysed by HPLC-DAD.

| Standard | Concentration (mg$_{standard}$/g$_{extract}$) |
|---|---|
| Caffeic acid | 0.15 |
| Catechin | 0.36 |
| Chlorogenic acid | 0.40 |
| Epicatechin | 0.32 |

The promising biological properties presented by the KP extract, in particular its antioxidant activity, make it an attractive natural ingredient to be incorporated into cosmetic formulations.

### 3.2. Characterisation of the Moisturising Creams

Taking into consideration the antioxidant properties presented by the KP extract, the possibility of using it as a natural additive in cosmetic products was evaluated. For that, four different moisturising creams were prepared. A formulation with no additives was used as the negative control (NC) and a formulation containing BHT, a synthetic antioxidant commonly used in the cosmetic industry, was used as the positive control (PC). Moreover, a formulation containing the extract (KPC) and another containing both BHT and KP extract (Mix) were also produced. Figure 2 shows the four moisturising creams after production. Homogeneous formulations were obtained with no significant visual differences in their colour.

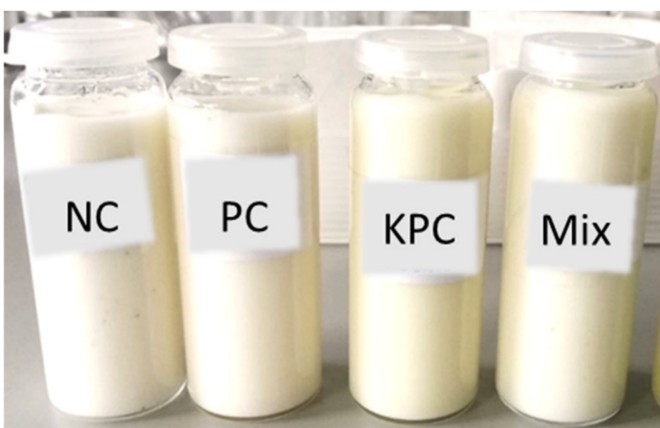

**Figure 2.** Moisturising cream formulations after production. NC—moisturising cream with no additives (negative control); PC—moisturising cream with a synthetic antioxidant (positive control); KPC—moisturising cream with kiwi peel extract; Mix—moisturising cream with BHT and kiwi peel extract.

The antioxidant properties of the prepared formulations were evaluated over two weeks. From Figure 3, it is possible to observe that the inhibition of the radicals was higher for ABTS than DPPH in all formulations. Also, all moisturising creams presented significantly higher inhibitions in comparison to the negative control in both DPPH and ABTS assays. Although this increase was superior for PC formulation, the KPC formulation also presented good antioxidant properties, particularly in the case of ABTS. The Mix formulation presented slightly better results than KPC, demonstrating that BHT and KP extract together can confer interesting antioxidant properties to moisturising creams. Finally, it was possible to analyse that the incorporation of the extract into the cosmetic formulation did not impair their antioxidant activity over time. In the DPPH assay (Figure 3A), this remained stable over two weeks (KPC formulation); in the ABTS assay (Figure 3B), although some variations were observed over time, these were identical for all formulations. The incorporation of other extracts from different by-products, namely onion and passion fruit peels, into cosmetic formulations was analysed in a different study. Those formulations also presented a significant increase in the antioxidant properties over two weeks [29]. These results demonstrate the potential of extracts containing phenolic compounds as antioxidants for cosmetic products.

To ensure that the addition of the extract to the moisturising cream did not affect its microbial profile, a microbial analysis was performed to verify the presence of microorganisms. Figure 4 shows that no coliform microorganisms (Figure 4A) or yeast and moulds (Figure 4B) were detected in any formulation one week after their production. These findings guaranteed that the incorporation of the KP extract did not alter the microbial safety of the product.

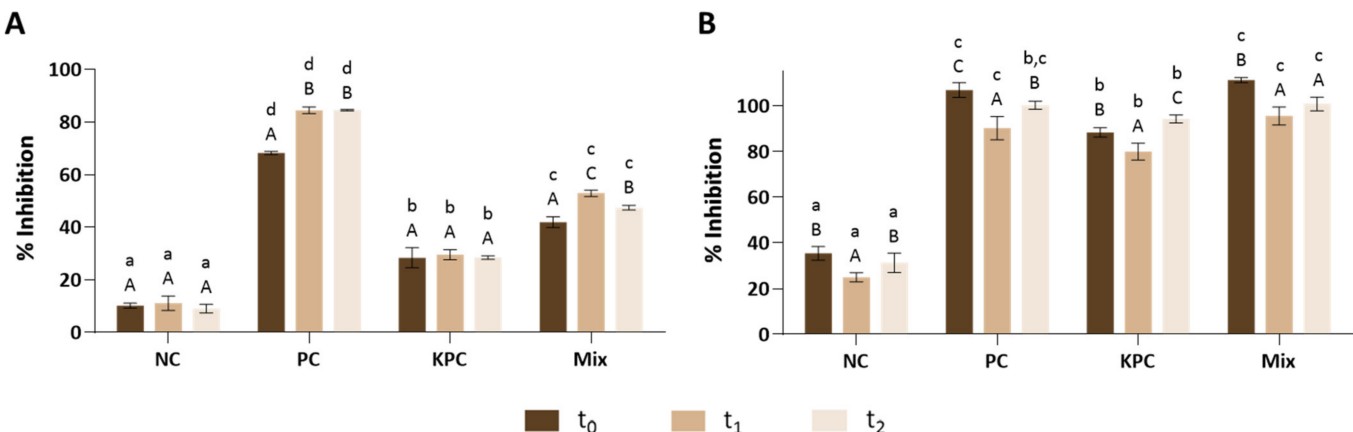

**Figure 3.** Antioxidant properties of the moisturising creams over two weeks for DPPH (**A**) and ABTS (**B**) assays. The analysis was performed after production ($t_0$), one week after production ($t_1$), and two weeks after production ($t_2$). NC—moisturising cream with no additives (negative control); PC—moisturising cream with a synthetic antioxidant (positive control); KPC—moisturising cream with kiwi peel extract; Mix—moisturising cream with BHT and kiwi peel extract. The results are expressed as means ± standard deviations of 3 independent measurements. Different lowercase letters (a–d) represent statistically different values ($p < 0.05$) for different formulations at the same timepoint. Different capital letters (A–C) represent statistically different values ($p < 0.05$) for the same formulation at different timepoints.

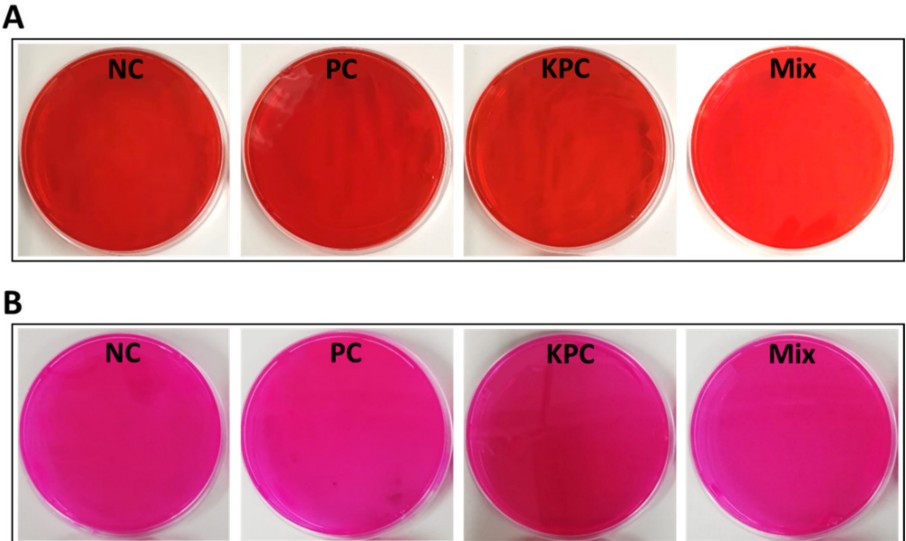

**Figure 4.** Microbial analysis of the moisturising cream formulations one week after production in a medium selective to coliform microorganisms, Lauryl Sulphate Agar (**A**), and a medium selective to yeast and moulds, Rose Bengal Chloramphenicol Agar (**B**). NC—moisturising cream with no additives (negative control); PC—moisturising cream with a synthetic antioxidant (positive control); KPC—moisturising cream with kiwi peel extract; Mix—moisturising cream with BHT and kiwi peel extract.

Finally, the stability of the formulations was investigated. For that, the physical stability and the pH of the samples were analysed over two weeks. The accelerated physical stability was assessed by submitting the formulations to intense centrifugation forces and checking for visual physical alterations. The appearance of the formulations before and after centrifugation can be examined in Figure 5. No phase separation or colour changes were observed right after the production of the moisturising creams (Figure 5A) nor two weeks after production (Figure 5B). These results demonstrated that the incorporation

of the extract obtained from KP did not impair the physical stability of the moisturising creams over the two weeks of the analysis. Other studies have already demonstrated that the incorporation of extracts from agro-industrial waste does not pose a negative impact on the physical stability of cosmetic products [29,31].

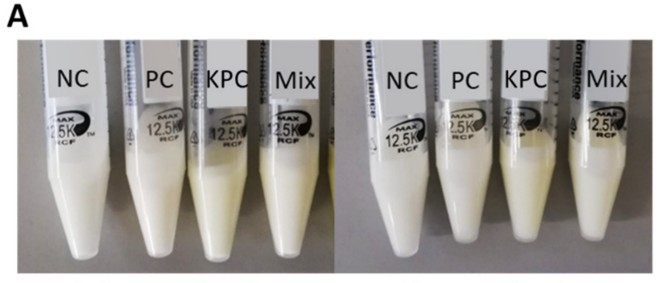
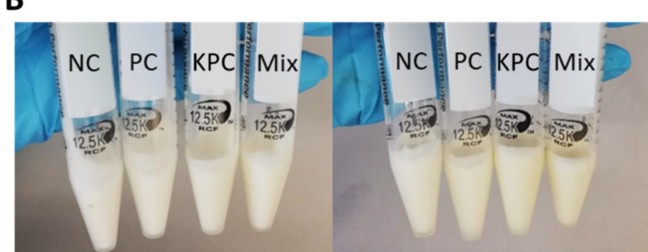

**Figure 5.** Moisturising cream formulations appearance before and after centrifugation after production (**A**) and two weeks after production (**B**). NC—moisturising cream with no additives (negative control); PC—moisturising cream with a synthetic antioxidant (positive control); KPC—moisturising cream with kiwi peel extract; Mix—moisturising cream with BHT and kiwi peel extract.

The pH value of the formulations was also determined over two weeks. This is an important aspect because the normal pH range of the skin is between 4.5 and 6, therefore, cosmetic products should present a pH value within that range [32]. According to Table 5, only KPC formulations presented a pH value slightly lower than the pH range of the skin. The addition of the KP extract to the moisturising cream reduced the pH of the samples (both KPC and Mix). This was expected due to the slightly acidic nature of the phenolic groups [33] and was also observed in another study where phenolic-rich extracts were added to cosmetic formulations [29]. Finally, a slight decrease was observed in the pH of PC and KPC formulations after two weeks, but the values remained between 4 and 6, typical for the skin.

**Table 5.** Moisturising cream formulations pH over two weeks.

| Formulation | pH | |
|:---:|:---:|:---:|
| | $t_0$ | $t_2$ |
| NC | 6.04 ± 0.03 [c] | 5.96 ± 0.02 [c] |
| PC | 6.05 ± 0.05 [c,B] | 5.93 ± 0.08 [c,A] |
| KPC | 4.11 ± 0.05 [a,B] | 4.02 ± 0.02 [a,A] |
| Mix | 4.46 ± 0.01 [b] | 4.53 ± 0.01 [b] |

NC—moisturising cream with no additives (negative control); PC—moisturising cream with a synthetic antioxidant (positive control); KPC—moisturising cream with kiwi peel extract; Mix—moisturising cream with BHT and kiwi peel extract. The analysis was performed after production ($t_0$) and two weeks after production ($t_2$). The results are expressed as means ± standard deviations of 3 independent measurements. Different lowercase letters (a–c) represent statistically different values ($p < 0.05$) in the same column. Different capital letters (A, B) represent statistically different values ($p < 0.05$) in the same line.

The evaluation of the antioxidant properties and stability of the moisturising creams revealed that KP extracts have the potential to be used alone or in combination with a synthetic antioxidant to produce sustainable value-added cosmetic products.

## 4. Conclusions

The present work aimed to investigate the possibility of using kiwi by-products, in particular kiwi peels, for the valorisation of cosmetic formulations. The characterisation of the kiwi peel extract revealed that this presented antioxidant and antibacterial properties, which can be explained by the phenolic compounds present, namely caffeic acid, catechin, chlorogenic acid and epicatechin. The incorporation of the obtained extract into moisturising cream formulations increased their antioxidant activity and did not impair

their microbiological safety. Also, the moisturising creams were shown to be stable for two weeks and their pH values were considered safe for a product intended for topical applications. Overall, the results demonstrated that it is possible to obtain a value-added cosmetic product by the incorporation of kiwi peel extracts. This work is a proof of concept of a possible application to valorise extracts from kiwi by-products, applying the principles of sustainability and circular economy to the cosmetic industry. Despite the promising results presented, other studies need to be performed to evaluate the safety as well as other properties of the final product. Cytotoxicity studies of the extract must be performed to confirm its security for human applications. The stability of cosmetic formulations should be analysed for longer periods of time, ideally six months, and their moisturising properties must also be assessed. Also, a comparison with a commercial moisturising cream would help clarify the benefits of the product proposed in this work compared to the cosmetics already available on the market. Finally, it would be of utmost importance to perform a sustainability assessment, analysing factors such as carbon footprint, resource consumption, and end-of-life disposal, to clearly understand the impact of the production of this novel cosmetic product.

**Supplementary Materials:** The following supporting information can be downloaded at https://www.mdpi.com/article/10.3390/su151914059/s1. Figure S1: Chromatograms of the standards obtained by HPLC-DAD analysis; Figure S2: Chromatograms of the extract obtained by HPLC-DAD analysis.

**Author Contributions:** Conceptualization, L.S.; methodology, S.M.G., R.M. and L.S.; validation, S.M.G. and R.M.; formal analysis, S.M.G.; investigation, S.M.G. and R.M.; resources, L.S.; data curation, S.M.G.; writing—original draft preparation, S.M.G. and R.M.; writing—review and editing, L.S.; visualization, S.M.G. and L.S.; supervision, L.S.; project administration, L.S.; funding acquisition, L.S. All authors have read and agreed to the published version of the manuscript.

**Funding:** This work was financially supported by LA/P/0045/2020 (ALiCE), UIDB/00511/2020 and UIDP/00511/2020 (LEPABE), funded by national funds through FCT/MCTES (PIDDAC).

**Institutional Review Board Statement:** Not applicable.

**Informed Consent Statement:** Not applicable.

**Data Availability Statement:** The data presented in this study are available on request from the corresponding author.

**Conflicts of Interest:** The authors declare no conflict of interest.

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
