# Peer review of "Sustainable Cosmetics: Valorisation of Kiwi (Actinidia deliciosa) By-Products by Their Incorporation into a Moisturising Cream"

_sustainability, doi:10.3390/su151914059_

Round 1

Reviewer 1 Report

Dear Author;

The manuscript's topic is timely and will be of interest to the journal readers.

Also, the manuscript is very well written, and the ideas flow logically. The review of the literature is thorough, so the reader is given an adequate background about the topic. Also, " Sustainable cosmetics: valorisation of kiwi by-products by their incorporation into a moisturising cream" has been assessed by me. Although it is of interest, we are able to consider it for publication in its current form. I have raised several points which we believe would improve the manuscript and may allow the current version to be published in the Sustainability.

Author Response

NOTE: The line numbers indicated in the answers represent the number of the line of the manuscript where the modification was performed when the modifications are visible.

1: Introduction: pls add origin of kiwi plant? And some more detail about the plant

Answer: This information was added to the Introduction section, as suggested (lines 49-61).

2: The kiwi peels were collected from Viseu, Portugal… pls add GPS location and which season?

Answer: The information was added to the manuscript, as suggested (lines 125-126).

3: Add further studies too?

Answer: The information about future work was added to the Conclusion section (lines 448-457).

Reviewer 2 Report

The manuscript is entitled " Sustainable cosmetics: valorisation of kiwi by-products by their incorporation into a moisturising cream” The objective of this study was to examine the potential of KP extract in terms of its antioxidant and antibacterial capabilities, as well as its influence when integrated into cosmetics, namely a moisturizing cream. The potential of utilizing this extract as an alternative or to reduce the use of synthetic preservatives in cosmetics products was also examined. In my perspective, it is essential to thoroughly consider and examine the following aspects:

In the introduction, the toxicity or safety data of the extract should be added, especially in human subjects or at least in animal models, because the kiwi peel extract is not in the CosIng Cosmetics Ingredients database.

The more kiwi information has to be provided in detail, such as the scientific name of the kiwi, the harvesting periods, the cultivar type, the sample deposition, or the voucher number,

In line 128, how do you know the particle size?

The scientific name must be italic, and the strain of microorganism also has to be provided.

In line 194, the conditions for the store sample, such as temperature, humanity, and light, must be provided.

Please provide the reasons why the authors kept the sample for only 2 weeks before analyzing the properties. Normally, the acceleration condition for the stability test is 6 months.

The positive controls for antioxidants and antibacterial have to be provided.

In the study, only a simple crude extract was prepared, but the authors claim that it was a phenolic-rich extract. The references that define the meaning of phenolic-rich have to be provided.

The HPLC chromatogram of the extract and standard should be provided.

The aim of the study incorporated cosmetic products, namely a moisturizing cream, so the moisturizing properties had to be investigated.

Please provide the reasons why the authors study antioxidant and antibacterial properties for moisturizing cream in terms of humectant or emollient effects.

Moderate editing of English language required

Author Response

NOTE: The line numbers indicated in the answers represent the number of the line of the manuscript where the modification was performed when the modifications are visible.

In the introduction, the toxicity or safety data of the extract should be added, especially in human subjects or at least in animal models, because the kiwi peel extract is not in the CosIng Cosmetics Ingredients database.

Answer: Thank you for your comment. To the best of the authors' knowledge, no toxicity analyses were performed in animal models or humans with kiwi peels extract. However, few studies have evaluated its cytotoxicity. A brief paragraph about this topic was added to the Introduction section (lines 109-113).

The more kiwi information has to be provided in detail, such as the scientific name of the kiwi, the harvesting periods, the cultivar type, the sample deposition, or the voucher number.

Answer: Thank you for your comment. The scientific name was added to the Title and the harvesting period was added to the Materials and Methods section (lines 125-126).

In line 128, how do you know the particle size?

Answer: The particle size is already indicated in the manuscript (line 156).

The scientific name must be italic, and the strain of microorganism also has to be provided.

Answer: The name of the microorganisms (italics) was checked and corrected in the entire manuscript and the strain of each microorganism was added to the Materials and Methods section (lines 185-186).

In line 194, the conditions for the store sample, such as temperature, humanity, and light, must be provided.

Answer: The information was added to the manuscript, as suggested (lines 233-234).

Please provide the reasons why the authors kept the sample for only 2 weeks before analyzing the properties. Normally, the acceleration condition for the stability test is 6 months.

Answer: Thank you for your comment. This work intended to be a proof of concept for the use of kiwi peels in cosmetic formulations and therefore, the properties of the formulations were analysed for a short period of time. Nevertheless, the authors understand the importance of analysing the stability of the formulations for longer periods of time and intend to perform those studies in future work, as well as other analyses to provide a more complete characterisation of the final product. A paragraph about future work was added to the Conclusion section (lines 448-457).

The positive controls for antioxidants and antibacterial have to be provided.

Answer: The information was added to the manuscript, as suggested.

In the study, only a simple crude extract was prepared, but the authors claim that it was a phenolic-rich extract. The references that define the meaning of phenolic-rich have to be provided.

Answer: Thank you for your comment. The authors used the term “phenolic-rich” since the phenolic compounds were the main compounds of interest in this investigation and the extraction method chosen was selected to optimise the extraction and this type of compounds. Although no treatment was performed on the crude extract to concentrate these compounds and other compounds are also present, the authors believe that the extract obtained still is an extract rich in phenolic compounds.

The HPLC chromatogram of the extract and standard should be provided.

Answer: Thank you for your comment. Although the authors understand the importance of providing the HPLC chromatograms, the chromatogram of the extract is quite complex as well as its analysis. To identify and quantify the different compounds in the extract, and since a diode array detector (DAD) was used, the analysis of the chromatogram was performed at different wavelengths and scales. Therefore, the authors believe that a picture of the chromatogram would be difficult to read and would not give all the information to the reader. 

The aim of the study incorporated cosmetic products, namely a moisturizing cream, so the moisturizing properties had to be investigated.

Answer: Thank you for your comment. The primary focus of this work was to assess the possibility of incorporating kiwi peels extracts into cosmetic formulations (using a moisturizing cream as an example) to give a new life to the bioactive compounds present in this by-product. The phenolic compounds, known for their antioxidant properties, are some of the bioactive compounds present in the kiwi peels; therefore, the studies performed in this study were more focused on the antioxidant properties of the formulations. However, the authors understand the importance of analysing the moisturising properties of the cosmetic products produced and will take this into consideration in future work. A paragraph about future work was added to the Conclusion section (lines 448-457).

Please provide the reasons why the authors study antioxidant and antibacterial properties for moisturizing cream in terms of humectant or emollient effects.

Answer: Emollients are oily substances which are, consequently, prone to lipid oxidation. Antioxidant compounds can play an important role in preventing/delaying this oxidation, maintaining the properties of the cosmetic product for prolonged periods of time. Therefore, the authors thought that it was of utmost importance to study the antioxidant properties of moisturising creams. On the other hand, humectants are hydrophilic compounds that attract and retain water. The moist environment is favourable to the growth of bacteria, which can impair the microbiological safety of the product. Therefore, the authors also analysed the antibacterial properties of the moisturising creams.

Reviewer 3 Report

General Summary (sustainability-2587054):

This manuscript entitled “Sustainable cosmetics: valorisation of kiwi by-products by their incorporation into a moisturising cream” by Gomes et al. deals with the biological properties and phenolic composition evaluation of kiwi fruit peels. This work is interesting as it provides significant insight into the management of fruit peel waste by unveiling their biochemical constituents and their further utilization in the cosmetic industry. However, the manuscript requires some improvements before it is finally considered for publication by the Sustainability Journal. My specific comments are given below:

1.      Please add the scientific name and botanical authority of the kiwi fruit (mention variety if applicable) in the title.

2.      Your manuscript should be thoroughly scrutinized by a native English writer to correct syntax and grammatical errors.

3.      The abstract section should be redrafted. No major numerical results are given.

4.      It would be better to write “agro-industrial wastes” instead of “agro-industrial by-products”.

5.      Graphical abstract: Please mention what are possible environmental and socioeconomic impacts caused by kiwi peel waste.

6.      Antioxidant capacity and antibacterial capacity should be changed to activities.

7.      Figure 1: Please add the role of each constituent provided both in humans and plants. This can be done by adding extra text near each component.

8.      Methods: Please correct the molecular formula of chemicals and reagents (sub and super scripts are wrongly written).

9.      Please write the name and model of the instrument used for wavelength measurement.

10.   While the paper presents the properties and performance of the Kiwi by-product moisturizing cream, it lacks a direct comparison with conventional moisturizing creams available in the market.

11.   Although the authors touch on the environmental benefits of by-product utilization, a more comprehensive sustainability assessment could strengthen their argument. Factors such as the overall carbon footprint, resource consumption, and end-of-life disposal of the new cream should be addressed.

12.   Several scientific names are not correctly written, please check the whole manuscript thoroughly.

13.   The conclusion section should be shortened and avoid repeating your abstract or discussion. Please write the overall outcome, limitations, and future directions. Expanding the study's scope to explore the use of by-products from other sources could enhance the paper's applicability and relevance to the cosmetics industry.

Your manuscript should be thoroughly scrutinized by a native English writer to correct syntax and grammatical errors.

Round 2

Reviewer 2 Report

Several aspects of the manuscript have been clarified, while others remain ambiguous.

The authors state that "To the best of the authors' knowledge, no toxicity analyses were performed in animal models or humans with kiwi peels extract. However, few studies have evaluated its cytotoxicity. A brief paragraph about this topic was added to the Introduction section (lines 109-113)", but some studies indicated that kiwi allergy has been on the rise in recent years (https://www.ncbi.nlm.nih.gov/pmc/articles/PMC10346195/). The authors should discuss this point. 

 Please provide the voucher number, collection year, and month of plant samples.

Please explain why the authors chose sorbic acid, which does not exhibit activity, as a positive control.

The authors stated that "The authors used the term "phenolic-rich" since the phenolic compounds were the main compounds of interest in this investigation and the extraction method chosen was selected to optimise the extraction and this type of compounds. Although no treatment was performed on the crude extract to concentrate these compounds and other compounds are also present, the authors believe that the extract obtained still is an extract rich in phenolic compounds". But the total concentration of phenolic acid that was present in the MS was only 1.23 mg/g extract (caffeic acid, catechin, chlorogenic acid, and epicatechin). Please provide a more specific reason or evidence.

The authors stated that "Although the authors understand the importance of providing the HPLC chromatograms, the chromatogram of the extract is quite complex as well as its analysis. To identify and quantify the different compounds in the extract, and since a diode array detector (DAD) was used, the analysis of the chromatogram was performed at different wavelengths and scales. Therefore, the authors believe that a picture of the chromatogram would be difficult to read and would not give all the information to the reader". Although it might be difficult to read, this is an important point, as the authors state. The authors should present the chromatogram and clearly explain it to increase the quality of the manuscript.

Author Response

NOTE: The line numbers indicated in the answers represent the number of the line of the manuscript where the modification was performed when the modifications are visible.

Several aspects of the manuscript have been clarified, while others remain ambiguous.

The authors state that "To the best of the authors' knowledge, no toxicity analyses were performed in animal models or humans with kiwi peels extract. However, few studies have evaluated its cytotoxicity. A brief paragraph about this topic was added to the Introduction section (lines 109-113)", but some studies indicated that kiwi allergy has been on the rise in recent years (https://www.ncbi.nlm.nih.gov/pmc/articles/PMC10346195/). The authors should discuss this point.

Answer: Thank you for your comment. As suggested, a brief sentence was added to the Introduction section of the manuscript  (lines 110-112).

Please provide the voucher number, collection year, and month of plant samples.

Answer: The authors added information about the collection year and month of the samples, as suggested (lines 126-127). However, the authors do not have the information about the voucher number of the plant.

Please explain why the authors chose sorbic acid, which does not exhibit activity, as a positive control.

Answer: The authors chose sorbic acid since it is one of the preservatives used in the cosmetic industry.

The authors stated that "The authors used the term "phenolic-rich" since the phenolic compounds were the main compounds of interest in this investigation and the extraction method chosen was selected to optimise the extraction and this type of compounds. Although no treatment was performed on the crude extract to concentrate these compounds and other compounds are also present, the authors believe that the extract obtained still is an extract rich in phenolic compounds". But the total concentration of phenolic acid that was present in the MS was only 1.23 mg/g extract (caffeic acid, catechin, chlorogenic acid, and epicatechin). Please provide a more specific reason or evidence.

Answer: Thank you for your comment. The authors revised the terminologies and opted to modify the term “phenolic-rich extract” to “extract” in the entire manuscript.

The authors stated that "Although the authors understand the importance of providing the HPLC chromatograms, the chromatogram of the extract is quite complex as well as its analysis. To identify and quantify the different compounds in the extract, and since a diode array detector (DAD) was used, the analysis of the chromatogram was performed at different wavelengths and scales. Therefore, the authors believe that a picture of the chromatogram would be difficult to read and would not give all the information to the reader". Although it might be difficult to read, this is an important point, as the authors state. The authors should present the chromatogram and clearly explain it to increase the quality of the manuscript.

Answer: Thank you for your comment. The authors added the images of the chromatograms in Supplementary Materials.

Reviewer 3 Report

All issues have been resolved. I suggest acceptance.

Author Response

Thank you for your comment.

Round 3

Reviewer 2 Report

-